# Cost-effectiveness of insulin pumps compared with multiple daily injections both provided with structured education for adults with type 1 diabetes: a health economic analysis of the Relative Effectiveness of Pumps over Structured Education (REPOSE) randomised controlled trial

Daniel John Pollard,[1] Alan Brennan,[1] Simon Dixon,[1] Norman Waugh,[2] Jackie Elliott,[3] Simon Heller,[3] Ellen Lee,[4] Michael Campbell,[1] Hasan Basarir,[5] David White,[4] On behalf of the REPOSE group

For numbered affiliations see end of article.

**Correspondence to**
Daniel John Pollard;
d.j.pollard@sheffield.ac.uk

## ABSTRACT

**Objectives** To assess the long-term cost-effectiveness of insulin pumps and Dose Adjustment for Normal Eating (pumps+DAFNE) compared with multiple daily insulin injections and DAFNE (MDI+DAFNE) for adults with type 1 diabetes mellitus (T1DM) in the UK.

**Methods** We undertook a cost–utility analysis using the Sheffield Type 1 Diabetes Policy Model and data from the Relative Effectiveness of Pumps over Structured Education (REPOSE) trial to estimate the lifetime incidence of diabetic complications, intervention-based resource use and associated effects on costs and quality-adjusted life years (QALYs). All economic analyses took a National Health Service and personal social services perspective and discounted costs and QALYs at 3.5% per annum. A probabilistic sensitivity analysis was performed on the base case. Further uncertainties in the cost of pumps and the evidence used to inform the model were explored using scenario analyses.

**Setting** Eight diabetes centres in England and Scotland.

**Participants** Adults with T1DM who were eligible to receive a structured education course and did not have a strong clinical indication or a preference for a pump.

**Intervention** Pumps+DAFNE.

**Comparator** MDI+DAFNE.

**Main outcome measures** Incremental costs, incremental QALYs gained and incremental cost-effectiveness ratios (ICERs).

**Results** Compared with MDI+DAFNE, pumps+DAFNE was associated with an incremental discounted lifetime cost of +£18 853 (95% CI £6175 to £31 645) and a gain in discounted lifetime QALYs of +0.13 (95% CI −0.70 to +0.96). The base case mean ICER was £142 195 per QALY gained. The probability of pump+DAFNE being cost-effective using a cost-effectiveness threshold of £20 000 per QALY gained was 14.0%. All scenario and subgroup

## Strengths and limitations of this study

► This study is the first cost-effectiveness analysis to consider the use of insulin pumps for adults with type 1 diabetes who are eligible to receive structured education course but not an insulin pump using current UK guidance.

► The Relative Effectiveness of Pumps over Structured Education (REPOSE) trial was the first trial to address the question of whether insulin pumps were a clinically effective treatment option in this population.

► An existing validated model of type 1 diabetes and its complications, developed during a previous National Institute for Health Research funded programme grant, was adapted and updated with evidence from the REPOSE trial and more recent evidence on clinical effectiveness, utilities and costs.

► We considered a UK healthcare perspective to estimate the key drivers of decision uncertainty.

► The main limitations of the study are that the only available evidence in long-term trends in HbA1c (the key clinical outcome) was from several observational studies, the analysis of diabetic ketoacidosis was based on self-reported information rather than data from inpatient admissions and that the trial only included one type of insulin pump.

analyses examined indicated that the ICER was unlikely to fall below £30 000 per QALY gained.

**Conclusions** Our analysis of the REPOSE data suggests that routine use of pumps in adults without an immediate clinical need for a pump, as identified by National Institute for Health and Care Excellence, would not be cost-effective.

**Trial registration number** ISRCTN61215213.

## INTRODUCTION

People with type 1 diabetes mellitus (T1DM) are unable to produce insulin due to autoimmune destruction of their insulin-secreting beta cells. Those affected have to inject insulin to prevent diabetic ketoacidosis (DKA) in the short term, and in the long term, to prevent microvascular and macrovascular disease. Insulin is generally administered by intermittent subcutaneous injection with the dose adjusted according to eating, physical activity and current blood glucose levels. Therapy is designed to keep blood glucose as close to normal as possible, to prevent both microvascular complications and to reduce the risk of macrovascular disease.[1] A further aim of treatment is to achieve as good a quality of life as possible, particularly since self-management of the condition is challenging and arduous, demanding the implementation of complex skills.

Historically, insulin was given twice a day, often as premixed insulin, but such an approach imposes a rigid lifestyle on people with T1DM and makes it difficult to maintain blood glucose levels close to normal. Most individuals require intensive insulin therapy to maintain tight glycaemic control. This approach and its integration within a flexible lifestyle is promoted in structured training courses, such as the Dose Adjustment For Normal Eating (DAFNE)[2] course and others.[3] The principles of this multiple daily injections (MDI) via subcutaneous injection approach involves the use of quick-acting insulin injected before eating (with the dose calculated according to the amount of carbohydrate eaten) combined with long-acting background 'basal' insulin, usually given twice daily, to control blood glucose in between meals.

As an alternative to MDI, insulin can be administered by an insulin pump system in which insulin is delivered throughout the day using a small, portable pump, connected by a flexible plastic tube to a subcutaneous cannula. This technology is relatively expensive compared with MDI. Current National Institute for Health and Care Excellence (NICE) guidelines recommend that all adults with T1DM in the UK receive a structured education course of proven benefit at a clinically appropriate time.[4] Insulin pumps are a treatment option for adults with T1DM who either have a HbA1c above 69 mmol/mol (8.5%) or experience disabling hypoglycaemia.[5] Insulin pumps do not replace the need for education, and it is currently recommended that specialist teams should provide structured education programmes and advice on diet, lifestyle and exercise for people using an insulin pump.[5]

Use of pumps in T1DM varies substantially between countries. In England, 11.7% of people with T1DM are estimated to use a pump, which compares to around 40% in the USA.[6 7] It has been proposed that insulin pumps are underused in the UK and that the glycaemic control of adults with T1DM could be improved if pumps were used more widely.[8] Two NICE appraisals found that there was insufficient evidence on insulin pump therapy compared with MDI in which adults with T1DM used both long-acting and short-acting insulins.[9 10] Furthermore, a recent observational study of pumps and MDI suggested that much of the previous benefit attributed to pumps may be due to the additional education that pumps users require when initiating therapy.[11] Therefore, uncertainty remains as to the clinical benefit and cost-effectiveness of pumps as a treatment option for adults with T1DM.

The Relative Effectiveness of Pumps over Structured Education (REPOSE) trial was conducted to assess if pumps offered any additional benefit compared with MDI for the treatment of adults with T1DM in the UK who are eligible to attend a structured education course but do not have an immediate clinical need for a pump. Participants in both trial arms received a DAFNE course, which taught MDI users to use long-acting and short-acting insulins appropriately. In this paper, we present the cost-effectiveness analysis of pumps+DAFNE compared with MDI+DAFNE for adults in the UK with T1DM who are eligible to receive a structured education course but did not have a clinical need for the immediate commencement of pump therapy.

## METHODS

### Economic evaluation methods

The health economic analysis followed the prespecified health economic analysis plan, which was outlined in the trial protocol paper.[12] Two approaches were undertaken to assess the cost-effectiveness of pumps: an economic evaluation alongside the clinical trial (EEACT) and a long-term economic model to assess the lifetime outcomes. In the EEACT, data on the costs and quality-adjusted life years (QALYs) were obtained from data collected in the REPOSE trial, whereas the long-term model estimated the lifetime costs and QALYs based mainly on the biomedical outcomes collected in the REPOSE trial. All modelling analyses took a lifetime time horizon, and all EEACT analyses took a 2-year (trial duration) time horizon. The prespecified primary analysis relates to the lifetime modelling, and as such, are reported in this paper. The methods and results of the EEACT are reported in online supplementary material A. In line with NICE guidance, all economic analyses took a National Health Service and personal social services perspective, and costs and QALY outcomes were discounted at 3.5%.[13]

### The economic model

The Sheffield Type 1 Diabetes Policy Model version 1.3.2, henceforth 'the model', is an individual level simulation model used to estimate the lifetime costs and QALYs associated with pump+DAFNE and MDI+DAFNE. The most recently published version of the model[14] was updated using data collected in the REPOSE trial at 12-month and 24-month follow-up. Specifically, the level of HbA1c, risk of severe hypoglycaemia, risk of DKA and the probability that an individual would switch insulin delivery mechanism and costs related to the intervention (insulin and MDI consumables, diabetes related contacts with healthcare professionals, insulin pumps and the associated consumables) were updated. Relevant literature

also informed the changes in HbA1c beyond the 2-year trial duration and the probability of death from end-stage renal disease (see online supplementary material B, page 2) in both arms. The model examines disease progression over a lifetime using an annual time cycle. An individual's HbA1c determines their risk of progression for all diabetic complications in the model, which include: nephropathy, neuropathy, retinopathy, macular oedema, myocardial infarction, stroke, heart failure, angina and severe hypoglycaemia and DKA. A higher HbA1c increases the risk of progression for all complications in the model. Individuals in the model are at risk of death from the incidence of: nephropathy, myocardial infarction, stroke, heart failure, angina and all-cause mortality. HbA1c indirectly effects mortality in the model, as the probability death does not differ by HbA1c; however, the risk of experiencing these events is higher for someone with a higher HbA1c. The model attaches utilities to health states and costs to events, allowing the calculation of costs and QALYs over a lifetime. Full details of how the model calculates the incidence of diabetes complications is provided in Heller *et al* (p. 104–106) and Thokala *et al.*[14 15] The incremental cost-effectiveness ratio (ICER) calculated was compared with the £20 000 to £30 000 per QALY gained threshold used by NICE.[13] In each model run, the life course of 5000 individuals was simulated. This number of simulated individuals was considered to be sufficiently robust for decision making (see online supplementary material B, page 1). All model analyses were conducted using SIMUL8 professional 2010.[16]

## The clinical data

The details of the methods used in the REPOSE trial have been reported elsewhere.[12] The eligibility criteria of the trial included adults with T1DM if they were eligible to receive a structured education course, did not have a clinical indication to receive a pump immediately, as determined by the investigator, or did not have a strong preference to receive a pump.[12] The REPOSE trial was conducted in eight centres in England and Scotland, involving 267 individuals. Out of these 267 individuals, 260 had HbA1c data for at least one postbaseline follow-up visit (intention to treat (ITT) population) and 236 adhered to their randomised treatment (per protocol population). Self-reported participant information, which includes the number of DKA events, EuroQol EQ-5D-3L, 12 item short form survey (SF-12) and resource use were collected at baseline, 6 months, 1 year and 2 years after attending DAFNE training. HbA1c, the primary endpoint, was measured at baseline and at these follow-up appointments. All inpatient admissions, the incidence of severe hypoglycaemia and whether an individual switched insulin delivery method were collected on an ongoing basis. Information was collected on the staff time spent delivering precourse fitting sessions for individuals allocated to receive a pump. The cost of insulin pumps and associated consumables in routine practice, and the insulin pump consumables used by the REPOSE participants, were collected from seven of the eight trial centres.

## The simulation cohort

To obtain the simulation cohort, participants' REPOSE data (n=260) were sampled with replacement 5000 times.[17] This created a simulation cohort of 5000 individuals for whom some would have missing baseline characteristics. Each individual from REPOSE was included in the simulation cohort a median of 19 times (IQR 16–22). To obtain data values for these missing variables in the simulation cohort, two imputation procedures were used. A truncated regression procedure for missing continuous variables (baseline: total cholesterol, high-density lipoprotein (HDL) cholesterol, systolic blood pressure and cost of insulin used in the year prior to baseline), in which variables were limited to positive values, and a Poisson procedure for missing categorical variables (gender). In the truncated regression imputation procedure, all characteristics with complete data in the REPOSE trial population were used as predictive covariates. In the Poisson imputation procedure, all characteristics with complete data in the REPOSE trial population and the imputed data for the continuous variables were used as predictive covariates. In both imputation procedures, a single imputation was used. As this was preformed in the simulation cohort rather than the REPOSE data, the replications of the individuals in the simulation cohort had different data values if they had missing baseline data from the trial. Identical individuals were simulated in both model arms. A summary of the baseline characteristics and completeness of data for the 260 REPOSE participants in the ITT population and the 5000 individuals in the simulation cohort are given in table 1.

## The clinical effectiveness data

Statistical analyses of the REPOSE trial data were conducted to estimate treatment switching, HbA1c, severe hypoglycaemia and DKA. The main clinical results paper is available; however, it differs in this paper as it focuses on statistical in those people with a HbA1c greater than or equal to 58 mmol/mol (7.5%).[18] Information on long-term changes in HbA1c was obtained from the available literature and the REPOSE trial data. Unless otherwise stated, all statistical analyses were conducted using STATA V.13.1.[19] Full details on the statistical methods used, the results and how the analyses were incorporated into the model are provided in online supplementary material B.

To estimate the incidence of treatment switching in the first and second year of the model, a time-to-event analysis was conducted using treatment switching as the event of interest. Kaplan-Meier curves were plotted, and parametric survival curves were fitted to the Kaplan-Meier curves.[20 21] Separate parametric survival models were fitted to individuals randomised to pump+DAFNE and MDI+DAFNE. The goodness of fit of the parametric survival curves was assessed using the Akaike information criterion, Bayesian information criterion and visual assessment of the fit of the parametric curves to the Kaplan-Meier curves at 1 and 2 years. As treatment switching was included in the model, it is important to

**Table 1** The baseline characteristics of REPOSE participants and the simulated cohort

| Characteristic | REPOSE trial population (n=260) | Simulated cohort (n=5000) |
|---|---|---|
| Continuous variables mean (SD) (% of individuals with data prior to imputation) | | |
| Baseline HbA1c (mmol/mol) | 76.0 (18.6) (100) | 75.9 (18.2) (100) |
| Baseline HbA1c (%) | 9.1 (1.7) (100) | 9.1 (1.7) (100) |
| Age (years) | 40.4 (13.4) (100) | 40.4 (13.3) (100) |
| Diabetes duration (years) | 18.0 (12.5) (100) | 18.0 (12.3) (100) |
| Triglycerides (mmol/mol) | 1.4 (1.0) (100) | 1.4 (0.9) (100) |
| Total cholesterol (mmol/mol) | 4.9 (0.9) (99.6) | 4.9 (0.9) (99.7) |
| HDL cholesterol (mmol/mol) | 1.6 (0.4) (96.5) | 1.6 (0.4) (96.4) |
| LDL cholesterol (mmol/mol) | 2.8 (0.9) (96.2) | 2.7 (0.9) (96.1) |
| Systolic blood pressure | 131.4 (16.4) (98.8) | 131.3 (16.0) (98.9) |
| Baseline cost of insulin | £357.24 (147.65) (94.8) | £360.39 (157.92) (98.4) |
| Baseline cost of diabetes-related contacts | £561.61 (885.92) (100) | £571.63 (928.92) (100) |
| Categorical variables n/N (percentage) (% of individuals prior to imputation) | | |
| Gender | | |
| Female | 104/260 (40.0) (40.0) | 1990/5000 (39.8) (39.3) |
| Male | 152/260 (58.5) (58.5) | 2950/5000 (59.0) (59.3) |
| Missing | 4/260 (1.5) (1.5) | 0/5000 (0.0) (1.4) |
| Physical activity | | |
| Low | 67/260 (25.8) (25.8) | 1266/5000 (25.3) (25.3) |
| Medium | 128/260 (49.2) (49.2) | 2471/5000 (49.4) (49.4) |
| High | 65/260 (25.0) (25.0) | 1263/5000 (25.3) (25.3) |
| Smoking status | | |
| Current | 50/260 (19.2) (19.2) | 951/5000 (19.2) (19.2) |
| Former | 67/260 (25.8) (25.8) | 1325/5000 (26.3) (26.3) |
| Never | 143/260 (55.0) (55.0) | 2724/5000 (54.5) (54.5) |
| Race | | |
| White | 258/260 (99.2) (99.2) | 4959/5000 (99.2) (99.2) |
| Black | 2/260 (0.8) (0.8) | 41/5000 (0.8) (0.8) |
| Nephropathy | | |
| No complications | 239/260 (91.9) (91.9) | 4600/5000 (92.0) (92.0) |
| Microalbuminuria | 13/260 (5.0) (5.0) | 234/5000 (4.7) (4.7) |
| Macroalbuminuria | 7/260 (2.7) (2.7) | 152/5000 (3.0) (3.0) |
| Dialysis or transplant | 1/260 (0.4) (0.4) | 14/5000 (0.4) (0.4) |
| Neuropathy | | |
| No complications | 238/260 (91.5) (91.5) | 4599/5000 (92.0) (92.0) |
| Neuropathy or ulcers | 22/260 (8.5) (8.5) | 401/5000 (8.0) (8.0) |
| Retinopathy | | |
| No complications | 145/260 (55.8) (55.8) | 28/5000 (56.0) (56.0) |
| Background diabetic retinopathy | 91/260 (35.0) (35.0) | 1740/5000 (34.8) (34.8) |
| Proliferative diabetic retinopathy | 24/260 (9.2) (9.2) | 465/5000 (9.3) (9.3) |
| Myocardial infarction | | |
| No complications | 255/260 (98.1) (98.1) | 4896/5000 (97.9) (97.9) |
| Myocardial infarction | 5/260 (1.9) (1.9) | 104/5000 (2.1) (2.1) |
| Stroke | | |
| No complications | 259/260 (99.6) (99.6) | 4983/5000 (99.7) (99.7) |
| Stroke | 1/260 (0.4) (0.4) | 17/5000 (0.3) (0.3) |

| Table 1 Continued | | |
|---|---|---|
| Characteristic | REPOSE trial population (n=260) | Simulated cohort (n=5000) |
| Heart failure | | |
| No complications | 259/260 (99.6) (99.6) | 4934/5000 (99.6) (99.6) |
| Heart failure | 1/260 (0.4) (0.4) | 18/5000 (0.4) (0.4) |
| Angina | | |
| No complications | 257/260 (98.9) (98.9) | 4934/5000 (98.7) (98.7) |
| Angina | 3/260 (1.2) (1.2) | 66/5000 (1.3) (1.3) |

HDL, high-density lipoprotein; LDL, low-density lipoprotein; REPOSE, Relative Effectiveness of Pumps over Structured Education.

adjust the modelled treatment effectiveness and cost of treatment for those individuals who switch. If individuals switched treatment, then their HbA1c was assumed to change as though they had been allocated to the other arm. No explicit effect of switching on the incidence rate ratio (IRR) associated with the model arms for either severe hypoglycaemia or DKA was included. However, due to the change in HbA1c associated with switching, individuals who switched from pump to MDI were at a lower risk of severe hypoglycaemia and at a higher risk of DKA. The opposite was true for those individuals who switched from MDI to pump. In the base case analysis, the HbA1c effect of pump+DAFNE compared with MDI+DAFNE was estimated in the per protocol population, as the people who switched treatment were excluded from this population.

To estimate each individual's HbA1c in the model, a beta regression was fitted to the HbA1c data collected in REPOSE for all individuals in the per-protocol population at the 1-year and 2-year follow-up.[22] Bounds were placed on the beta regression so that HbA1c was between the clinically plausible bounds of 29 mmol/mol (4.8%) and 201 mmol/mol (20.5%), which were provided by two clinical experts in the REPOSE trial management group. Missing data were observed for HbA1c values in the per-protocol population at 6 months (2.1% missing), 1 year (4.2% missing) and 2 years (4.2% missing). Full details on the imputation procedure used to account for the missing data and the specification of the beta regression are provided in online supplementary material B (see p. 8). The effects in a beta regression are not easily interpretable by themselves; however, information can be obtained on the direction of effect.

The expert opinion of clinical members involved in the literature review of clinical studies for the REPOSE trial was sought to identify studies with long-term data on HbA1c for the type of pumps used in the trial and people who used MDI after DAFNE. Three articles on the long-term trends in HbA1c for pump users[23–25] and two articles on the long-term trends in HbA1c for MDI users post-DAFNE[26 27] were used to estimate yearly changes in HbA1c for each model arm. The average annual increase in HbA1c was calculated for each study by calculating yearly increase in HbA1c between the lowest observed HbA1c value and the last follow-up point in the study where the sample size was greater than one quarter of the

initial sample size. An SE could not be directly calculated, as we did not have access to the patient level data from the studies. Instead, data on the SD of the change in HbA1c between the 1-year and 2-year follow-ups specific to each REPOSE trial arm were used to inform the uncertainty in this parameter. These SDs were divided by the square root of the combined sample size of the studies used to inform the mean effect to estimate SE for this parameter.

Negative binomial regressions were used to estimate the risk of severe hypoglycaemia and DKA in the ITT population of REPOSE using the Zellig package in R V.3.2.0. Negative binomial regressions were fitted separately to severe hypoglycaemia and DKA events and to the first and second years of follow-up data. The regressions fitted to the second year data were used to estimate the incidence of severe hypoglycaemia and DKA in all model time cycles, except for the first time cycle. This assumption was based on clinical expert opinion, which was confirmed by the trial management group, that the first 6 months of using a pump was associated with a learning period, and after this time, the incidence of severe hypoglycaemia and DKA would decrease.

## Costs and utilities

The costs and health state utility values used for each health state in the model are provided in table 2 and table 3, respectively. All costs were reported in 2013/2014 prices. Costs sourced from evidence in previous years were inflated to 2013/2014 prices using the hospital and community health services pay and prices index.[28] The cost of insulin and MDI consumables used, cost of diabetes related contacts with healthcare professionals and cost of the pump intervention costs were estimated separately for both year 1 and year 2 of the REPOSE trial. The unit cost of insulin and MDI consumables used was obtained from the British National Formulary and Health & Social Care Information Centre, the unit cost of diabetes related contacts were obtained from NHS reference costs and unit cost of an insulin pump (with its associated consumables) was obtained from a survey of the REPOSE trial sites.[29–31] Table 2 part A shows the six regression formulae that estimate an individual's annual cost for each of these components as a function of: their baseline HbA1c, which treatment they start the year on (MDI or pump), whether they switch treatment and also

**Table 2** The health state cost parameters used in the Sheffield Type 1 Diabetes Policy Model

**Part A: seemingly unrelated regression functions for estimated costs in year 1 and ongoing based on REPOSE trial data (multivariate normal distributions*)**

| | Annual cost of insulin and MDI consumables (year 1) | Annual cost of insulin and MDI consumables (ongoing) | Annual cost of DRC (year 1) | Annual cost of DRC (ongoing) | Annual cost of insulin pump and associated consumables (year 1) | Annual cost of insulin pump and associated consumables (ongoing) | Sources |
|---|---|---|---|---|---|---|---|
| Multiplier for the baseline DRC cost ($\psi_1$) | – | – | +0.11 | +0.03 | – | – | REPOSE trial data in year 1. (year 1 costs). REPOSE trial data in year 2. (ongoing costs). NHS reference costs.[30] Pump costing survey. |
| Multiplier for the baseline insulin cost ($\psi_2$) | +0.97 | +1.04 | – | – | – | – | |
| Multiplier for the baseline HbA1c (DCCT % scale) ($\psi_3$) | +5.08 | +12.81 | –21.66 | +12.15 | – | – | |
| Receiving pump at the start of the year ($\psi_4$) | –517.91 | –527.64 | +129.08 | +88.99 | +2056.11 | +2050.99 | |
| Switch from pump to MDI ($\psi_5$) | +554.47 | +153.35 | +280.16 | –47.10 | –1143.68 | –905.03 | |
| Switch from MDI to pump ($\psi_6$) | 0† | –353.27 | +733.95 | +201.93 | +804.57 | +1134.27 | |
| Constant ($\psi_0$) | +381.77 | +324.53 | +415.46 | +299.80 | 0.00 | 0.00 | |

**Part B: costs of DAFNE fixed parameters**

| Health state | Mean cost (£) | SE | Source | Health state | Mean cost (£) | SE | Source |
|---|---|---|---|---|---|---|---|
| Cost of a DAFNE course | 363.10 | None | DAFNE Fact Sheet Six[37] | Cost of precourse fitting session for pump users | 28.82 | None | DAFNE Fact Sheet Six[37] and REPOSE trial data |

**Part C: costs of adverse events, comorbidities and complications Gamma distributions**

| Health state | Mean Cost (£) | SE | Source | Health state | Mean cost (£) | SE | Source |
|---|---|---|---|---|---|---|---|
| Adverse events | | | | | | | |
| Hypoglycaemia | 187 | 18.69 | Heller et al[14] | DKA with hospitalisation | 1399 | 140 | NHS reference costs[38] |
| Nephropathy | | | | | | | |
| Ongoing yearly cost of microalbuminuria | 36 | 3.56 | BNF[39] and McEwan et al[40] | Ongoing yearly cost of microalbuminuria ongoing | 36 | 3.56 | BNF[39] and McEwan et al[40] |
| Ongoing yearly cost of ESRD | 24436 | 2444 | NHS reference costs[38] | Death due to ESRD | 0 | 0 | Assumed equal to zero |
| Neuropathy | | | | | | | |
| Clinically confirmed neuropathy | 271 | 27.14 | Currie et al[41] | Clinical neuropathy | 271 | 27.14 | Currie et al[41] |
| Diabetic foot syndrome | 2848 | 285 | NHS Reference costs[38] | PAD with amputation (year 1) | 7221 | 722 | NHS Reference costs[38] |
| Ongoing yearly cost of PAD with amputation | 439 | 43.93 | McEwan et al[40] | | | | |

Continued

**Table 2** Continued

Part C: costs of adverse events, comorbidities and complications Gamma distributions

| Health state | Mean Cost (£) | SE | Source | Health state | Mean cost (£) | SE | Source |
|---|---|---|---|---|---|---|---|
| **Retinopathy** | | | | | | | |
| Background retinopathy | 145 | 14.47 | McEwan et al[40] | Proliferative retinopathy | 661 | 66.11 | McEwan et al[40] |
| Macular oedema (year 1) | 5710 | 571.0 | NICE,[42] BNF[43] | Macular oedema (year 2) | 3416 | 341.6 | NICE,[42] BNF[43] |
| Macular oedema (year 3) | 2562 | 256.2 | NICE,[42] BNF[43] | Macular oedema (ongoing) | 277 | 27.7 | NICE,[42] BNF[43] |
| Blindness (year 1) | 1584 | 158 | Clarke et al[44] | Blindness (ongoing) | 519 | 51.88 | Clarke et al[44] |
| **Cardiovascular** | | | | | | | |
| First MI (year 1) | 6788 | 679 | Clarke et al[44] | Second MI (year 1) | 6788 | 679 | Clarke et al[44] |
| Final MI (year 1) | 6788 | 679 | Clarke et al[44] | Ongoing yearly cost of an MI | 904 | 90.43 | Clarke et al[44] |
| Fatal MI | 2101 | 210 | Clarke et al[44] | Heart Failure (year 1) | 3818 | 382 | Clarke et al[44] |
| Heart failure (ongoing) | 1173 | 117 | Clarke et al[44] | Fatal HF | 3818 | 382 | Clarke et al[44] |
| First stroke (year 1) | 4361 | 436 | Clarke et al[44] | Second stroke | 4361 | 436 | Clarke et al[44] |
| Fatal stroke | 5684 | 568.45 | Clarke et al[44] | First stroke (ongoing) | 559 | 55.90 | Clarke et al[44] |
| Angina (year 1) | 3397 | 340 | Clarke et al[44] | Angina (ongoing) | 951 | 95.09 | Clarke et al[44] |

*The variance covariance matrices used to parameterise the multivariate normal distribution are provided in online supplementary material B tables 4, 7 and 12.

†This variable was included in the original regressions; however, the model would converge when this covariate was included, so this parameters is taken to be a zero value in the total cost formula.

−This value was not included as a covariate in the regression formula and is taken to be a zero value in the total cost formula.

The cost for each total cost in part A is calculated using the following formula:

Total cost=$\beta_0$+$\beta_1$*individual's baseline diabetes-related contact cost+$\beta_2$*individual's baseline insulin cost+$\beta_3$*individual's baseline HbA1c (DCCT % scale)+$\beta_4$*individual's treatment at the start of the year (1=pump, 0=MDI)+$\beta_5$*(1=switched from pump to MDI, 0=did not switch from pump to MDI)+$\beta_6$*(1=Switched from MDI to pump, 0=did not switch from MDI to pump).

BNF, British National Formulary; DAFNE, Dose Adjustment for Normal Eating; DCCT, Diabetes Control and Complications Trial; DKA, diabetic ketoacidosis; DRC, diabetes related contacts; ESRD, end-stage renal disease; HF, heart failure; MDI, multiple daily injections; MI, myocardial infarction; NHS, National Health Service; NICE, National Institute for Health and Care Excellence; PAD, peripheral arterial disease; pump, insulin pumps; REPOSE, Relative Effectiveness of Pumps over Structured Education.

**Table 3** The base case utility parameters

**Beta distribution**

| Health state for event | Utility | SE | *Alpha* | *Beta* | Source |
|---|---|---|---|---|---|
| Baseline utility value | | | | | |
| Male with type 1 diabetes and no complications | 0.866 | 0.010 | 947.79 | 146.90 | Peasgood et al[45] |

**Gamma distribution**

| | Disutility | SE | *Alpha* | *Beta* | Source |
|---|---|---|---|---|---|
| Complications or covariates | | | | | |
| Female with type 1 diabetes and no complications | 0.0236 | 0.008 | 8.70 | 0.003 | Peasgood et al[45]* |
| Adverse events‡ | | | | | |
| Severe hypoglycaemia | −0.002 | −0.002 | 1 | 0.002 | Peasgood et al[45] |
| Diabetic ketoacidosis | −0.0091 | −0.01 | 0.83 | 0.01 | Peasgood et al[45]* |
| Nephropathy§ | | | | | |
| Microalbuminuria | 0 | | | | Assumption |
| Microalbuminuria | −0.017 | 0.01 | 2.89 | 0.01 | Coffey et al[46] |
| ESRD | −0.078 | 0.026 | 9 | 0.01 | Coffey et al[46] |
| Neuropathy§ (applied to the history of events) | | | | | |
| Clinical neuropathy | −0.055 | 0.01 | 30.25 | 0.002 | Coffey et al[46] |
| Clinically confirmed neuropathy | −0.055 | 0.01 | 30.25 | 0.002 | Coffey et al[46] |
| Diabetic foot syndrome | −0.1042 | −0.119 | 0.77 | 0.14 | Peasgood et al[45] |
| PAD with amputation | −0.1172 | −0.055 | 4.54 | 0.03 | Peasgood et al[45]* |
| Retinopathy§ (applied to the history of events) | | | | | |
| Background retinopathy | −0.0544 | −0.023 | 5.59 | 0.01 | Peasgood et al[45] |
| Proliferative retinopathy | −0.0288 | −0.026 | 1.23 | 0.02 | Peasgood et al[45] |
| Blindness | −0.208 | 0.013 | 256 | 0.001 | Coffey et al[46] |
| Cardiovascular§ (applied to the history of events) | | | | | |
| MI (first year) | −0.065 | 0.03 | 4.69 | 0.01 | Alva et al[47] |
| MI (subsequent years) | −0.057 | 0.03 | 3.61 | 0.02 | Alva et al[47] |
| Heart failure | −0.101 | 0.032 | 9.96 | 0.010 | Alva et al[47] |
| Stroke | −0.165 | 0.035 | 22.22 | 0.007 | Alva et al[47] |
| Angina | −0.09 | 0.018 | 25 | 0.004 | Clarke et al[48]† |

*A parameter value was not available in the author's preferred statistical model.
†Value is presented in table 5 as ischaemic heart disease.
‡These disutilites are applied transiently to the number of these events in each year.
§These disutilites are applied to the history of ever having had one of these events.
ESRD, end-stage renal disease; MI, myocardial infarction; PAD, peripheral arterial disease.

the participants' self-reported use of insulin and level of contact with healthcare professionals prior to recruitment to the trial. Details on how these parameters were used to estimate the costs in the model are provided in table 2.

### Scenario and subgroup analyses

For the base case, a probabilistic sensitivity analysis (PSA) was conducted where each parameter was sampled from its probability distribution, the results were recorded, the process was repeated 500 times and the averages from these 500 model runs were reported. This number of PSA runs was considered to be sufficient to allow for robust decision making based on the

model results (see online supplementary material B, page 1). All statistical models, except for the risks of severe hypoglycaemia and DKA, for which the outcomes were directly simulated in R V.3.2.0, were included in the model using a multivariate normal distribution.[32] The long-term changes in HbA1c were assumed to be distributed using independent normal distributions for each model arm. Deterministic scenario analyses were undertaken to assess the robustness of the results. The key scenario analyses for the model include: uncertainty in the estimates of treatment effectiveness, the timing of HbA1c changes in the model, the utility decrement

for blindness $(-0.26)$[33] and the cost of insulin pumps and insulin pump consumables.

A two-way deterministic threshold analysis was conducted to assess the HbA1c reduction and/or annual cost reduction necessary to potentially make pumps cost-effective in the UK. All threshold analysis runs were conducted deterministically. In this analysis, each individual's HbA1c was estimated as though they received MDI. Then those individuals in the pump arm received a fixed change in HbA1c. This change in HbA1c was varied at 10 different values between $-3.3$ mmol/mol $(-0.3\%)$ and $-13.1$ mmol/mol $(-1.2\%)$, and annual cost of insulin pumps were also varied between 100% of the observed cost in the REPOSE trial and 50% of the observed cost.

Deterministic subgroup analyses were conducted in the following populations:

1. baseline HbA1c <69 mmol/mol (8.5%)
2. baseline HbA1c ≥69 mmol/mol (8.5%)
3. baseline HbA1c ≥58 mmol/mol (7.5%)
4. 69 mmol/mol (8.5%) >baseline HbA1c ≥58 mmol/mol (7.5%)
5. 80 mmol/mol (9.5%) >baseline HbA1c ≥69 mmol/mol (8.5%)
6. baseline HbA1c ≥80 mmol/mol (9.5%)
7. all individuals in the per protocol population.

The subgroup analyses were conducted by resampling the simulation cohort from the individuals who met each of these criteria in the ITT population.

## RESULTS
### Clinical effectiveness data
#### Treatment switching

The analysis of treatment switching in the REPOSE trial suggested that an exponential curve provided the best fit to the data for individuals receiving pump+DAFNE and a Weibull curve provided the best fit for individuals in the MDI+DAFNE arm.

#### HbA1c

The results of the analysis of HbA1c data suggested that pump+DAFNE compared with MDI+DAFNE was associated with statistically insignificant HbA1c reductions in both year 1 and in year 2. The analysis of the literature for long-term trends in HbA1c suggested that weighted average annual progression of HbA1c was +0.568 mmol/mol (+0.052%) per annum in the pump+DAFNE arm and +0.590 mmol/mol (+0.054%) per annum in the MDI+DAFNE arm. The estimated standard errors for the long-term trends in HbA1c were of 0.627 mmol/mol (0.040%) for MDI+DAFNE and 0.627 mmol/mol (0.057%) for pump+DAFNE.

These data produce a trace of HbA1c as in figure 1. The solid lines show the average HbA1c in the model for each year. The dotted lines show what the HbA1c would have been in each year, if a higher HbA1c did not have an indirect effect on the number of deaths in the model or treatment switching effects. The dashed line shows the number of people remaining alive in each year and is

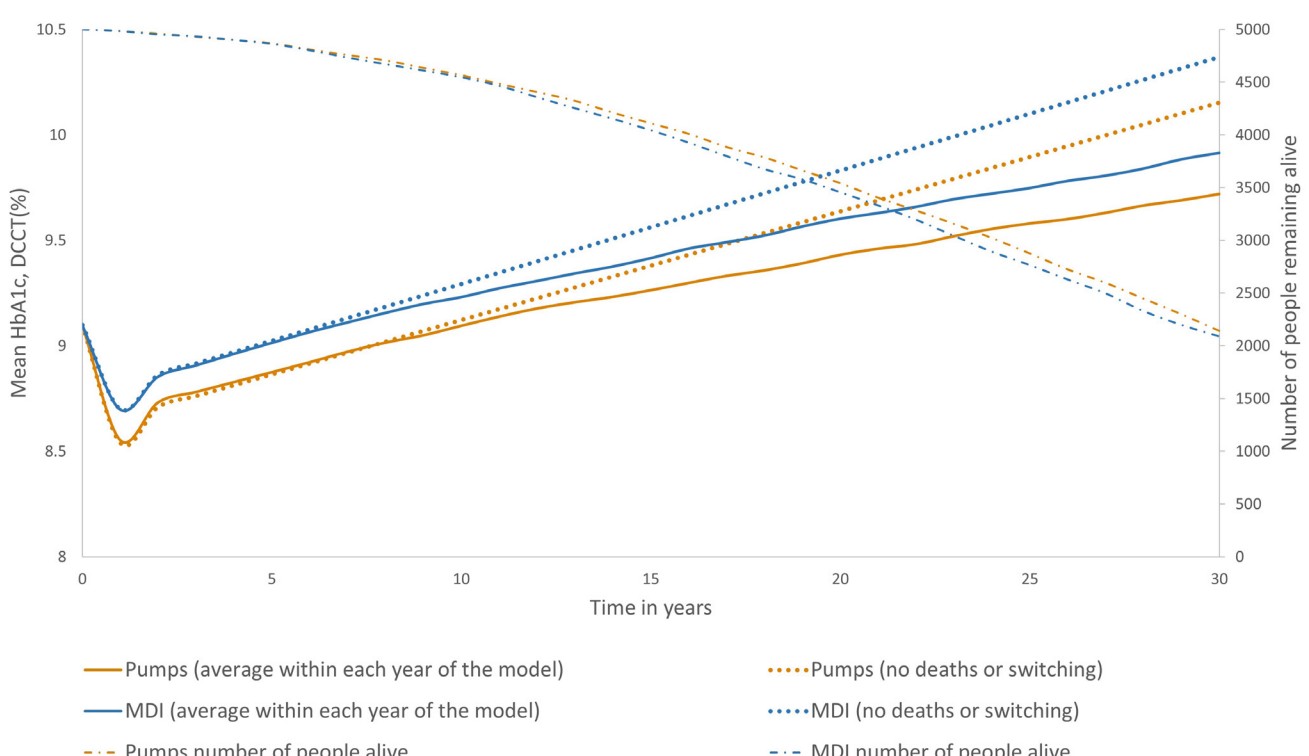

**Figure 1** The trace of: (A) the mean HbA1c in each year of the model, (B) the mean HbA1c ignoring the effects of death and treatment switching, (C) the number of people alive in each year of the model in both arms over 30 years. DCCT, Diabetes Control and Complications Trial; MDI, multiple daily injection.

plotted on the secondary axis. As expected from the base case clinical results, the MDI+DAFNE arm has a higher HbA1c than the pumps+DAFNE arm, and this effect slightly increases over time

### Severe hypoglycaemia and DKA

The results of the negative binomial models used to estimate the incidence of DKA indicated that pump+DAFNE compared with MDI+DAFNE was associated with more DKA events in the first year (IRR 1.40, 95% CI 0.55 to 3.58) but fewer events in year 2 (IRR 0.93, 95% CI 0.23 to 3.69). The results of the negative binomial models used to estimate the incidence of severe hypoglycaemia indicated that pump+DAFNE compared with MDI+DAFNE was associated with more severe hypoglycaemic events in year 1 (IRR 1.33, 95% CI 0.49 to 3.65) but fewer events in the second year (IRR 0.35, 95% CI 0.08 to 1.44).

### Base case results

Table 4 shows the base case results for the long-term cost-effectiveness analysis using the PSA. For the pump+DAFNE arm, the mean costs of the intervention are £42 124 discounted over the lifetime horizon, which compares with £19 829 for the MDI+DAFNE arm. This leads to an increase in intervention costs of £21 295 over a person's lifetime. The QALYs lived, prior to applying the utility decrements associated with diabetic complications, in the pump+DAFNE arm is 13.91 QALYs compared with 14.03 QALYs in the MDI+DAFNE arm, a difference of 0.11 QALYs. The differential incidence of adverse events in the pump+DAFNE arm compared with the MDI+DAFNE arm leads to a lower costs (−£371) and higher QALYs (+0.00) per person. The reduced incidence of diabetic complications in the pump+DAFNE arm leads to lower costs (−£2070) and more QALYs (+0.01) per person. The net incremental lifetime cost of pump versus MDI is therefore estimated as £18 853 (95% CI £6175 to £31,645) per person. The net incremental QALY gain per person is 0.13 (95% CI −0.70 to +0.96) QALYs per person. The ICER associated with pump+DAFNE was £142 195 per QALY gained. This is considerable above the £30 000 per QALY threshold, which is the higher limit at which NICE would usually consider interventions to be cost-effective.[13] Figure 2A shows that this result is subject to a small degree of uncertainty, as in 14.0% of the PSA runs, pump+DAFNE would be considered as being cost-effective using the lower end of the range used by NICE.

### Scenario and subgroup analyses

The results of 11 scenario analyses are presented in figure 2B. None of the scenario analyses showed pump+DAFNE to be cost-effective versus MDI+DAFNE. The lowest ICER found (£31 747 per QALY gained) was for scenario 3 in which yearly cost of pumps and consumables was halved.

The results of the two-way price and effectiveness threshold analysis are given in table 5. When the annual pump cost is assumed to be 100% of the prices observed

in REPOSE, then the analysis shows that the reduction in HbA1c (for pump compared with MDI) would need to be 12.0 mmol/mol (1.1%) or more, for pump to have an ICER below £20 000 per QALY gained. When the annual cost is 25% lower, then a HbA1c reduction of more than 7.7 mmol/mol (0.7%) would be needed to have an ICER below £20 000 per QALY gained. When the annual cost is halved, then a HbA1c reduction of 3.3 mmol/mol (0.3%) would be sufficient to have an ICER below £20 000 per QALY gained.

The result of the subgroup analyses are presented in figure 2C. Similarly to the scenario analysis, no subgroup was identified in which pump+DAFNE would be likely to be considered more cost-effective than MDI+DAFNE. The most cost-effective subgroup was those individuals with a baseline HbA1c greater than or equal to 80 mmol/mol (9.5%), with an ICER of £96 394 per QALY gained versus MDI+DAFNE.

### DISCUSSION

This is the first study to examine the marginal benefits and cost-effectiveness of insulin delivered using pumps over MDI when both groups have had structured education, in patients without an immediate clinical need for an insulin pump, as recommended by NICE. Our findings show that providing pumps for this wider group of adults with T1DM would be unlikely to be considered cost-effective in the UK because the estimated ICER was just over £142 000, substantially higher than the £30 000 per QALY upper limit often used by UK decision makers. This finding is also valid in all scenario and subgroup analyses examined. The threshold analysis indicates that if new insulin pump technology is developed and the costs are similar to current insulin pumps, a trial would have to demonstrate a HbA1c reduction in the region of 11 mmol/mol (1.0%) to 13 mmol/mol (1.2%) compared with MDI+DAFNE for the new technology to be considered cost-effective in the UK for the population analysed in this study. Conversely, if insulin pump prices were halved, then reductions of only 3 mmol/mol compared with MDI are required.

The key strengths of this study are that it is based on a rigorously conducted cluster RCT with economic data directly collected during the study. Data completeness for the primary outcome was very high at 95%. The study uses an individual level simulation model of type 1 diabetes disease progression over a lifetime horizon. The study does have limitations in terms of the evidence used to inform the long-term changes in HbA1c, which were based on five observational studies with follow-up ranging from 3.7 to 10 years, rather than trials with long follow-up periods. Also, as REPOSE is the first study to assess the effectiveness of pumps+DAFNE, the long-term evidence was based on observational studies of pumps in which the education component is likely to have been different from the DAFNE structured education provided in the REPOSE trial. Our PSA has incorporated

**Table 4** Long-term cost-effectiveness analysis: base case results using probabilistic sensitivity analysis

|  | MDI+DAFNE | Pump+DAFNE | Incremental |
|---|---|---|---|
| **Mean lifetime discounted costs per person** | | | |
| Intervention costs | | | |
| Insulin and MDI consumables | £12 215 | £5476 | −£6740 |
| Diabetes-related contacts | £5023 | £6289 | £1266 |
| Insulin pumps and pump consumables | £2228 | £28 967 | £26 739 |
| DAFNE course | £363 | £392 | £29 |
| Subtotal intervention costs | £19 829 | £41 124 | £21 295 |
| Adverse event costs | | | |
| Severe hypoglycaemia | £133 | £41 | −£92 |
| Diabetic ketoacidosis | £1161 | £882 | −£279 |
| Subtotal cost of adverse events | £1294 | £922 | −£371 |
| Long-term complication costs | | | |
| Nephropathy | £40 786 | £38 853 | −£1933 |
| Neuropathy | £1859 | £1805 | −£53 |
| Retinopathy+macular oedema | £6365 | £6263 | −£102 |
| Myocardial infarction | £1838 | £1844 | £6 |
| Heart failure | £607 | £609 | £2 |
| Stroke | £253 | £254 | £0 |
| Angina | £1134 | £1143 | £8 |
| Total cost of long-term complications | £52 841 | £50 771 | −£2070 |
| Total costs | £73 964 | £92 817 | £18 853 (95% CI £6175 to £31 645) |
| **Mean undiscounted life years per person** | | | |
| Total life years | 28.3181 | 28.7999 | 0.3790 (95% CI −2.7392 to 3.3403) |
| **Mean discounted QALYs per person** | | | |
| QALYs lived (excluding decrements due to complications) | 13.9145 | 14.0292 | 0.1147 |
| QALYs lost due to adverse events | | | |
| Severe hypoglycaemia | −0.0014 | −0.0004 | 0.0009 |
| Diabetic ketoacidosis | −0.0075 | −0.0057 | 0.0018 |
| Subtotal QALYs due to adverse events | −0.0088 | −0.0061 | 0.0027 |
| QALYs lost due to complications | | | |
| Nephropathy | −0.1853 | −0.1792 | 0.0061 |
| Neuropathy | −0.3092 | −0.3010 | 0.0082 |
| Retinopathy+macular oedema | −0.3316 | −0.3293 | 0.0022 |
| Myocardial infarction | −0.0528 | −0.0532 | −0.0004 |
| Heart failure | −0.0385 | −0.0387 | −0.0002 |
| Stroke | −0.0343 | −0.0345 | −0.0002 |
| Angina | −0.0754 | −0.0761 | −0.0007 |
| Subtotal QALYs lost due to complications | −1.0271 | −1.0120 | 0.0152 |
| Total QALYs | 12.8785 | 13.0111 | 0.1326 (95% CI −0.7087 to 0.9623) |
| **Summary** | | | |
| Total mean discounted costs per person | £80 471 | £99 337 | £18 853 (95% CI £6175 to £31 645) |

Continued

**Table 4** Continued

|  | MDI+DAFNE | Pump+DAFNE | Incremental |
|---|---|---|---|
| Total mean undiscounted life years per person | 28.3181 | 28.7999 | 0.3790 (95% CI −2.7392 to 3.3403) |
| Total mean discounted QALYs per person | 12.8785 | 13.0111 | 0.1326 (95% CI −0.7087 to 0.9623) |
| ICER (£/QALY gained) |  |  | £142 195 |
| Probability that pump+DAFNE is cost-effective at a threshold of £20 000 per QALY gained |  |  | 14.0% |

DAFNE, Dose Adjustment for Normal Eating; ICER, incremental cost-effectiveness ratio; MDI, multiple daily injections; pump, insulin pumps; QALY, quality-adjusted life year.

uncertainty around these estimates. Second, the analysis of rates of DKA was based on self-reported occurrence of DKA in REPOSE (the same method as used in other recent trials of DAFNE Heller *et al*, chapter 8),[14] rather than corroborated data on inpatient admissions that had a smaller number of events and could not be analysed statistically. Third, only one pump type was assessed in the trial, and some caution may be needed when considering extrapolating results to modern pumps with autosuspend features. Fourth, the results of the threshold analysis used to determine what the effectiveness of pumps would need to be considered cost-effective should be interpreted with some degree of caution for two reasons. First, we used a fixed HbA1c effect for every individual, rather than a method that accounts for heterogeneous treatment responses, and second, we assume switchers from MDI to pump get the fixed HbA1c reduction and switchers from pump to MDI get the same fixed HbA1c increase. In an

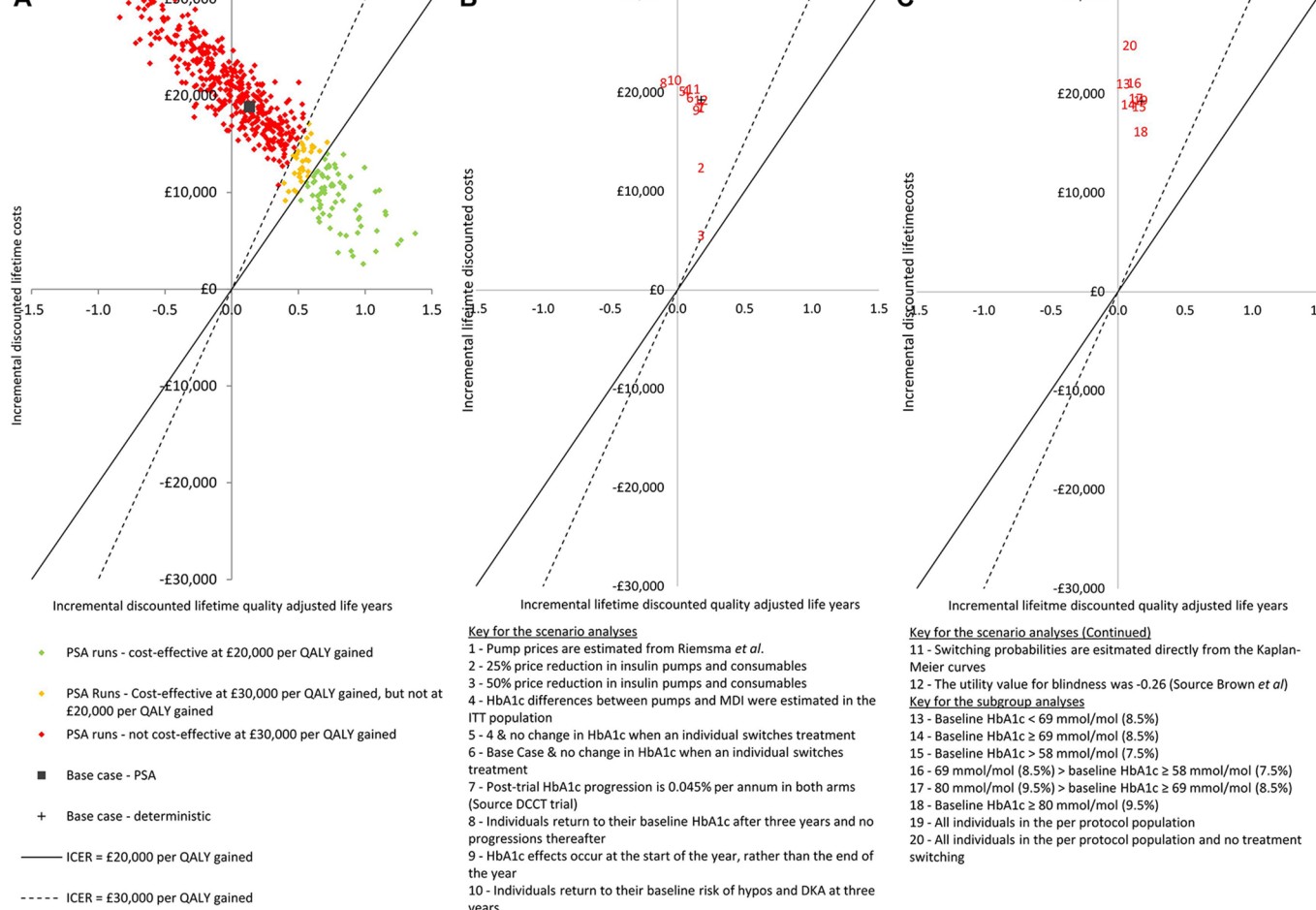

**Figure 2** The results of the economic analyses presented on the cost-effectiveness plane for: (A) the base case PSAs, (B) the results of the scenario analyses and (C) the results of the prespecified subgroup analyses. DCCT, Diabetes Control and Complications Trial; DKA, diabetic ketoacidosis; ICER, incremental cost-effectiveness ratio; MDI, multiple daily injection; PSA, probabilistic sensitivity analysis; QALYs, quality-adjusted life years.

**Table 5** The incremental cost-effectiveness ratio associated with insulin pumps for different HbA1c reductions (for all adults with type 1 diabetes mellitus) and annualised prices of insulin pumps and insulin pump consumables

| | 3.3 mmol/mol (0.3%) | 4.4 mmol/mol (0.4%) | 5.5 mmol/mol (0.5%) | 6.6 mmol/mol (0.6%) | 7.7 mmol/mol (0.7%) | 8.7 mmol/mol (0.8%) | 9.8 mmol/mol (0.9%) | 10.9 mmol/mol (1.0%) | 12.0 mmol/mol (1.1%) | 13.1 mmol/mol (1.2%) |
|---|---|---|---|---|---|---|---|---|---|---|
| £2060 (100%) | £85990 | £75710 | £64493 | £47667 | £34280 | £27951 | £25979 | £26258 | £18953 | £17610 |
| £1957 (95%) | £79106 | £69830 | £58901 | £43527 | £31132 | £25128 | £23494 | £23736 | £16730 | £15515 |
| £1854 (90%) | £72222 | £63950 | £53309 | £39387 | £27985 | £22306 | £21009 | £21213 | £14507 | £13419 |
| £1751 (85%) | £65338 | £58070 | £47717 | £35247 | £24837 | £19483 | £18524 | £18691 | £12284 | £11323 |
| £1648 (80%) | £58454 | £52189 | £42125 | £31107 | £21689 | £16661 | £16039 | £16169 | £10060 | £9227 |
| £1545 (75%) | £51570 | £46309 | £36533 | £26967 | £18542 | £13838 | £13554 | £13646 | £7837 | £7131 |
| £1442 (70%) | £44686 | £40429 | £30941 | £22827 | £15394 | £11016 | £11070 | £11124 | £5614 | £5036 |
| £1339 (65%) | £37802 | £34549 | £25349 | £18687 | £12246 | £8193 | £8585 | £8602 | £3391 | £2940 |
| £1236 (60%) | £30917 | £28668 | £19757 | £14547 | £9099 | £5370 | £6100 | £6079 | £1167 | £844 |
| £1133 (55%) | £24033 | £22788 | £14164 | £10407 | £5951 | £2548 | £3615 | £3557 | Dominates | Dominates |
| £1030 (50%) | £17149 | £16908 | £8572 | £6267 | £2803 | Dominates | £1130 | £1035 | Dominates | Dominates |

Red, the incremental cost-effectiveness ratio is above £30,000 per quality adjusted life year gained; Orange, the incremental cost-effectiveness ratio is between £30 000 and £20 000 per quality adjusted life year gained; Green, the incremental cost-effectiveness ratio is less than £20 000 per quality adjusted life year gained.

ITT analysis where switchers are included in their originally randomised arms, smaller HbA1c reductions than the ones in this analysis would likely lead to the same ICERs. Finally, as no one study could provide sufficient information on the disutility decrements associated with diabetic events for people with type 1 diabetes, the data come from disparate sources. This means that the magnitude of some events appears to be small compared with others. For example, end-stage renal disease has a utility decrement of −0.078, whereas heart failure has a higher decrement of −0.101.

A 2015 systematic review on the cost-effectiveness of insulin in various countries identified four studies from the UK, three of which presented an ICER.[34] The base case ICERs in these three studies were £11 461, £25 648 and £37 712, which indicate that in two out of the three studies that pumps had an ICER within/below the threshold range usually used by NICE to determine the cost-effectiveness of technologies in the UK. However, most of the previous cost-effectiveness analyses used a reduction in Hba1c of 10–13 mmol/mol (0.95%–1.2%) associated with pumps compared with MDI, based on the meta-analysis by Weissberg-Benchell et al[35] which included a mixture of RCTs and observational studies. This effect size is much larger than that observed in the REPOSE trial. Some of the effect size in the meta-analysis may be related to the education offered to people in the pump arms, which was not offered to people in the MDI arms and that MDI was not administered using both long-acting and short-acting analogue insulins in most studies included in the meta analysis. To our knowledge, REPOSE is the first large trial in adults with type one diabetes that provides evidence on pumps versus MDI (in which both long-acting and short-acting analogue insulins are used) that provides equivalent diabetes education to both trial arms. As such, this economic analysis is the first cost-effectiveness analysis relevant to assessing whether pumps should be offered at the point when adults with T1DM in the UK are eligible to receive structured education and do not have an immediate clinical need to receive a pump.

The key implication of this paper is that unless there is an immediate clinical need for using a pump, clinicians in the NHS should offer adults with T1DM a structured education course of proven benefit, prior to considering pump therapy. This is because when pumps+DAFNE is compared with MDI+DAFNE, the incremental health benefits are relatively small and the incremental lifetime costs are relatively high. This indicates that it would not be cost-effective to allow all adults with T1DM who are eligible for structured education, and have no clinical need for a pump, to also immediately receive an insulin pump.

A question not addressed by the REPOSE trial is whether pumps would be clinically effective and cost-effective in patients offered the technology at some point after having attended a structured education course. Generating this evidence would require a clinical study, which recruited participants who had previously received a structured education course of proven benefit and

randomising them to either continued MDI or pumps. While the current literature does provide some indication of which individuals may benefit from pumps, the clinical evidence informing these studies was not limited to those individuals who had first received structured education. This is important because the effectiveness of insulin pumps would be expected to differ in this group, with this being related to observed and unobserved patient characteristics. One particular hypothesis that we believe is worthwhile investigating is whether pumps should be used in the UK by those adults with T1DM who actively self-manage after attending a structured education course, but either have not achieved the NICE target HbA1c levels of less than or equal to 48 mmol/mol (6.5%) or who have problematic hypoglycaemia. Another important research question concerns the effectiveness and cost-effectiveness of support programmes posteducation, so that more adults with T1DM achieve glycaemic targets. The DAFNEplus NIHR programme grant, which will report in 2022, is developing and testing an adapted DAFNE course (based on previous research, behaviour change theory and technological support) and subsequent structured support to improve self-management and glucose control.[36]

In conclusion, the results indicate that it would not be cost-effective to offer pumps to all adults with T1DM in the UK, who are currently eligible to receive a structured education course and do not have an immediate clinical need for a pump in the UK. Use of MDI+DAFNE is estimated to represent a better use of NHS resources than immediate commencement on a pump. Further research is required to improve the glycaemic control of adults with T1DM in the UK.

**Author affiliations**
[1]School of Health and Related Research (ScHARR), University of Sheffield, Sheffield, UK
[2]Population Evidence and Technologies, Warwick Medical School, University of Warwick, Coventry, UK
[3]Academic Unit of Diabetes, Endocrinology and Metabolism, Department of Oncology and Metabolism, School of Medicine and Biomedical Sciences, University of Sheffield, Sheffield, UK
[4]Clinical Trials Research Unit, School of Health and Related Research (ScHARR), University of Sheffield, Sheffield, UK
[5]RTI Health Solutions, Manchester, UK

**Acknowledgements** We would like to thank Richard Jacques and Mónica Hernández Alava for their advice on how to conduct the statistical analysis on HbA1c and Peter Mansell for providing helpful comments on the full report.

**Collaborators** Simon Heller was the chief investigator. Norman Waugh was the deputy chief investigator. Stephanie Amiel, Mark Evans, Fiona Green, Peter Hammond, Alan Jaap, Brian Kennon, Robert Lindsay and Peter Mansell were site principal investigators and contributed to the study design and data interpretation. Jane Baillie, Anita Beckwith, Helen Brown, Karen Callaby, Katy Davenport, Sarah Donald, Jackie Elliott, Leila Faghahati, Sara Hartnell, Allison Housden, Kalbir Kaur Pabla, Nicola Croxon, Sheena Macdonald, Muna Mohammed, Vicky Steel, Katy Valentine, Pamela Young, Ann Boal, Patsy Clerkin, Lynn Doran, Joanne Flynn, Emma Gibb, Hilary Peddie, Bernie Quinn, Helen Rogers, Janice Shephard, Janet Carling, Ann Collins, Laura Dinning, Christine Hare, Joyce Lodge, Sutapa Ray, Debora Brown, Jenny Farmer, Alison Cox, Chris Cheyette, Pratik Choudhary, Linda East, June Ellul, Katherine Hunt, Kimberley Shaw, Ben Stothard, Lucy Diamond, Lindsay Aniello, Debbie Anderson, Kathy Cockerell, Vida Heaney, Alison Hutchison, Nicola Zammitt, Gayna Babington, Gail Bird, Janet Evans, Tasso Gazis, Nicola Maude, Karen Nunnick, Dawn Spick, Laura Fenn, Carla Gianfrancesco, Valerie Gordon, Linda Greaves, Susan Hudson, Valerie Naylor, Chloe Nisbet, Carolin Taylor, Karen Towse and Candice Ward contributed at sites to participant recruitment, intervention delivery and data collection. Cindy Cooper, Gemma Hackney, Diana Papaioannou, Emma Whatley and David White provided central trial management, oversight and monitoring. Mike Bradburn, Michael Campbell, Munya Dimairo and Ellen Lee contributed to the statistics. Hasan Basarir, Alan Brennan, Simon Dixon and Daniel Pollard contributed to the health economics. Nina Hallowell, Jackie Kirkham, Julia Lawton and David Rankin designed and undertook the qualitative work. Katharine Barnard led the quantitative psychosocial work. Timothy Chater and Kirsty Pemberton provided data management. Fiona Allsop and Lucy Carr provided central administration. Pamela Royle conducted literature searches and exploratory analyses. Gill Thompson and Sharon Walker provided central DAFNE support. Pauline Cowling conducted the fidelity assessment. Henry Smithson provided user representation on the management group.

**Contributors** DJP contributed to the design of all of the health economic analyses and conducted all statistical and modelling analyses used in this paper. AB oversaw the design and implementation of the modelling analyses. SD, oversaw the design and implementation of the economic analysis of the trial data and contributed to the estimation of the treatment costs used in the long-term economic analyses. AB, SD, NW, JE, SH and MC contributed to the design of the trial. NW, JE and SH provided clinical input into the design of the economic analyses. MC designed the statistical analysis plan. EL implemented the statistical analysis plan, parts of which were used in the economic evaluation. HB and DW contributed to data collection for the economic analyses. AB and HB designed and developed previous versions of the economic model used in these analyses. DJP wrote the first draft. All authors approved the final draft. DJP is the guarantor.

**Funding** This research was funded by the UK Health Technology Assessment Programme (project number 08/107/01). The research was sponsored by Sheffield Teaching Hospitals NHS Foundation Trust. The clinical sites taking part were: Cambridge University Hospitals NHS Foundation Trust, Dumfries and Galloway Royal Infirmary, NHS Greater Glasgow and Clyde, Harrogate and District NHS Foundation Trust, King's College Hospital NHS Foundation Trust, NHS Lothian, Nottingham University Hospitals NHS Trust and Sheffield Teaching Hospitals NHS Foundation Trust. We also acknowledge the financial support of the Research and Development Programmes of the Department of Health for England and the Scottish Health and Social Care Directorates, which supported the costs of consumables and of Medtronic UK Ltd, which provided the insulin pumps for the trial.

**Disclaimer** The views and opinions expressed herein are those of the authors and do not necessarily reflect those of the HTA, NIHR, NHS, the Department of Health or Medtronic UK Ltd.

**Competing interests** SH reports personal fees from Sanofi Aventis and personal fees and other from NovoNordisk and Eli Lilly, outside the submitted work. JE reports personal fees from Astra Zeneca, Merck Sharpe Dohme and Takeda, personal fees and non-financial support from Eli Lilly, Novonordisk and Sanofi, outside the submitted work. NW reports receiving two days of consultancy fees from Novo Nordisk on topics including alternatives to network meta-analysis, unrelated to REPOSE.

**Patient consent** Obtained.

**Ethics approval** Research Ethics Committee (REC) North West, Liverpool East.

**Provenance and peer review** Not commissioned; externally peer reviewed.

**Data sharing statement** Requests for patient level data and statistical code should be made to the corresponding author and will be considered by the REPOSE trial management group who, although specific consent for data sharing was not obtained, will release data on a case-by-case basis following the principles for sharing patient level data as described by Smith *et al* (2015). The presented data do not contain any direct identifiers; we will minimise indirect identifiers and remove free-text data to minimise the risk of identification.

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
