## [Reviewer comments · BMJ Open]

ARTICLE DETAILS

TITLE (PROVISIONAL)	The cost-effectiveness of insulin pumps compared to multiple daily injections, both provided with structured education, for adults with type 1 diabetes: a health economic analysis of the relative effectiveness of pumps over structured education (REPOSE) randomised controlled trial.
AUTHORS	Pollard, Daniel; Brennan, Alan; Dixon, Simon; Waugh, Norman; Elliott, Jackie; Heller, Simon; Lee, Ellen; Campbell, Michael; Basarir, Hasan; White, David

VERSION 1 – REVIEW

REVIEWER	Andrew J Palmer University of Tasmania, Australia
REVIEW RETURNED	10-Apr-2017

GENERAL COMMENTS	This is a well-written cost-effectiveness analysis of insulin pump in addition of education program versus education program and MDI in a UK setting. Thorough sensitivity/scenario analyses have been performed to assess the impact many different plausible assumptions regarding key parameters. The authors draw reasonable conclusions based on the analysis as presented. I have some minor comments and suggestions: 1) Abstract results – please report CIs for incremental costs and incremental QALYs.2) Limit reporting of all QALYs in abstract, main body of the manuscript, tables etc. to 2 decimal places (currently reported to 4 decimal places, which implies that the authors can predict QALYs to the nearest hour or so over patients' lifetime.....rather unrealistic...)3) A more detailed description of calculation of complication-specific mortality would be appreciated – currently it seems as if only ESRD-specific mortality is modelled apart from other causes of mortality. At least report that other disease-specific mortalities are modelled (e.g. from MI, stroke, heart failure, DKA, severe hypos) and whether or not HbA1c has a direct/indirect effect on these mortalities. If the model does not take into account HbA1c's impact on these forms of mortality, this is a major weakness and will lead to a higher ICER of pump vs MDI .4) There is a good description of the impact of switching therapies
--

	on HbA1c, but the impact of switching on hypo and ketoacidosis rates is not well described – please do so. Failure to adjust for these rates changing following treatment switch will impact on the model outcomes. 5) Can the authors assure the readers that the number of samples/model runs was sufficient to reach a stable ICER? 6) While the equations for modelling HbA1c are well described, it would make the paper a lot more transparent to the non-statistician if the authors show a trace of mean HbA1c over time for each comparator. Please provide this as well as for any other biomarkers simulated in the model. 7) Was there any differential impact on the comparators on BMI/weight? If so, how was this accounted for?
--	---

REVIEWER	Anil Gumber Faculty of Health and Wellbeing Sheffield Hallam University Sheffield (UK)
REVIEW RETURNED	24-Apr-2017

GENERAL COMMENTS	 • I have noticed at several places some grammar issues...in some places interpretation were confusing. See file attached • Further, the model is based on RESPOSE trial which has only two cases of Black patients and none from Asian ethnicity. The evidence suggests that prevalence rate of diabetes is much higher in Asian and Black compared to White adults. This is a major limitation of modelling exercise and thus not at all representative of patients' ethnicity. So I don't think of publishing this paper in its current form. The reviewer also provided a marked copy with additional comments. Please contact the publisher for full details.
---

REVIEWER	Danny Liew Monash University, Australia Previous and current involvement in clinical studies of diabetes care, including use of more intensive insulin therapy. Have received honoraria and research grants from pharmaceutical companies involved in therapies for diabetes, but not from any sponsors of insulin pumps.
REVIEW RETURNED	17-Oct-2017

GENERAL COMMENTS	This is a nicely-written paper which examines the cost-effectiveness of insulin pumps, compared to multiple daily injections, over a lifetime time horizon, based on the results of the REPOSE randomised clinical trial. The authors utilise a published model, the 'Sheffield Type 1 Diabetes Policy Model' to undertake the analyses, with updating of some key inputs. I was not familiar with the model prior to this review, but read the original publication with interest. The results of the cost-effectiveness analyses, including those of the probabilistic sensitivity analyses (PSA), were well presented and appropriately interpreted. Key limitations were identified, and seemingly would not have altered the conclusion of the study
--

	anyway. Ultimately, the study's applicability was limited to the patient sub-population represented by the REPOSE trial, and hence the extent to which this trial was reflective of the target clinical scenario was by and large the maximal extent to which the cost-effectiveness study was representative. Factors that may have diminished the validity of the study's finding beyond this included the potential non-validity of the Sheffield model and the inputs entered into it. However, as mentioned, the Sheffield model has been peer-reviewed and key inputs and assumptions seemed appropriately justified. I have just a few minor comments/queries:  1. In the main part of the manuscript, it would be helpful to describe in more detail how the Sheffield model associates HbA1C (and change in HbA1C) with risk of clinical outcomes, especially since this is the key mechanism through which benefits are achieved. 2. For the PSA, why were gamma distributions for disutilities associated with diabetes complications used? (Table 3) My understanding is that beta distributions (as was used for the baseline utility) are standard. 3. Were disutilities for severe hypoglycaemia and diabetic ketoacidosis applied transiently or permanently? It should be the former, as these conditions are transient. If so, how long were they applied for? 4. Deriving the multitude of disutility values and costs from disparate sources is a limitation that needs to be acknowledged. For example, the disutility for end-stage renal disease (-0.078) seems small compared to that for heart failure (-0.101).
--	--

REVIEWER	JC Gardiner Michigan State University, USA
REVIEW RETURNED	02-Nov-2017

GENERAL COMMENTS	Summary: The manuscript addresses the cost-effectiveness of insulin pumps (continuous subcutaneous insulin injections) compared to multiple daily injections in Type1 diabetes adult patients, both provided with a structured education on dose adjustment for normal eating The REPOSE trial of pumps with structured education compared to multiple daily injections (without the need for structured education) from the background, rationale, and primary data source for the project. The design is cohort simulation of patients following the Sheffield Type1 Diabetes Policy Model. Clinical endpoints are HbA1c, and risks of complications from diabetes. The likelihood of switching treatment delivery is addressed in the analysis. The document in its entirety is 130+ pages in length, including two separate Supplements (A, B), a published paper and protocol of REPOSE. I presume the main article covers the first 27 pages, through Figure 1. It includes Tables 1-5. The remainder of the massive document, includes supplementary details from REPOSE, a within-trial incremental cost-effectiveness analysis. Methods employed and model assumptions are described in great detail. All are appropriate. The cost-effectiveness analyses follow generally accepted guidelines for conduct and reporting of results—of course NICE (and
--

ISPOR) recommendations are followed.

Apart from the objectives of this specific study, this “project” is valuable for providing a sort of tutorial for delineating the compulsory elements for conduct, design and analysis of a cost-effectiveness study that has the backdrop of a substantive clinical trial with patient level data and resource utilization. To be useful in this regard, the online document must be assembled with links to assess table, figures and supplement. Up front a table of contents, links to chapters, etc is highly recommended.

Comments, edits and reflections

The bullets that follow are notes taken as I read through the document. Reference to pages are as in the PDF document on the top left or right hand corner.

- The comparison for the current article is insulin pumps vs multiple daily injections—ie, pumps+DAFNE, vs MDI +DAFNE
- After page 42 ‘pumps’ is replaced by CSII=continuous subcutaneous insulin injection
- REPOSE trial—of pumps with structured education compared to MDI (without the need for structured education). Both arms received DAFNE. Primary outcome HbA1c
- It would help the reader if a list of all abbreviations and acronyms placed upfront, perhaps just before the INTRODUCTION. At first mention later in the article the acronym is spelled-out and afterwards may be maintained
- Format the article as a mini-book, with index of contents, chapters and links to each chapter. I guess the Supplementary Materials A and B will be separate documents. (eg, The REPOSE Protocol from page 72)

Page2

- Economic Analyses: (1) alongside the clinical trial (EEACT), (2) long term modeling of outcomes and costs over the lifetime. Model for (2)—Sheffield Type1 Diabetes Policy Model (ver 1.3.2). The life course was simulated in 5,000 type 1 diabetes patients
- Please give a reference to SIMUL8 (2010) software, and for STATA

Current paper employs (2), and EEACT is in Supplementary materials A.

- Hard to locate Supplements A and B
- Basis/background/details for the model come from 2 articles, Heller et al [14] and Thokala et al [15]
- Cohort simulation Groot Koerkamp et al [16]. Please see if Groot Koerkamp et al (Medical Decision Making, JUL-AUG 2011) is a more appropriate reference
- Page7/line 125; ?replicated. Sentences 126-130 require clarification....[?] –single imputation was performed ..., by two procedures...(i) chained regression....., (ii) Poisson... What does ‘both sets’ of imputation models refer to?
- Page 7/line 136: suggest sub-rubrics for the three models that follow for (a) treatment switching, (b) HbA1c, (c) severe hypoglycaemia and DKA. They are separate outcomes in the model that are discussed next.
- Page7/line 143 ff: Time to treatment-switching as event. In Supplementary Materials B. Suggest -----survival curves estimated from parametric models were plotted against the nonparametric Kaplan-Meier curve.

There is only one KM curve (Figure 1/page 49 and Figure 2/page 50)) with 5 comparison parametric survival curves obtained from the Exponential, Weibull, Gompertz, log-logistic and log-normal distributions. For each these some sort of mixture distribution must

	be used because there are covariates in the parametric model (Tables 1-2, page 45-47) --- Is HbA1c (as covariate) in these model the single baseline value? In other words, it is not time-varying. HbA1c is recorded at baseline, 1 year and 2 years. However, many more assessments are likely available in this study. --- Only available covariate data were used in the time-to-event (treatment switching) estimations. So, mention to operational sample size (n=xxx) in the legend on the tables.  • Table 1-2 (pages 45-47): Presumably these are output by STATA. The shape/scale parameter names such “gamma”, “sigma” parameters, could be confusing unless we know the parameterization used. The survival functions (not all) are of the form $S(t) = \exp(-\theta t^\gamma)$ for scale θ and shape γ. For goodness-of-fit, criteria such as Kolmogorov-Smirnov (KS), Anderson-Darling (AD), Cramer von-Mises (CvM) should be added to Table 3. Where statistics involve the log-likelihood, $-2 \log L$ should include all estimated constants and based on the un-logged response • Page 51, Table 3 begs an explanation. Rubrics for row/column misaligned • Back to Page 8/para 1: This is a model for HbA1c to aid in imputation. So, drop “in the model” Page3  • Page 8 line 16: supplementary material (pg8) is actually pg9. In Supplementary Material: on this page (page 52), the beta regression model uses the logit link for the mean $E(Y) = \mu = \text{xx}$ and the variance is $\text{Var}(Y) = \phi \mu^{-xxx}$. The dispersion is on the variance (not on the mean as stated). Some software use ϕ^{-1} in place of ϕ. In tables that follow, --- Does “beta scale” mean “logit scale”? --- Natural logarithm of phi. I guess it is the parameterization above --- Patients were nested in one of 7 clinic sites. Any concerns or issues on analyses if clustering should be accounted for? --- Page 56 (pg13 in Supplement): Is Table 9 to correct reference? --- Page 59 (pg16 in Supplement): Table 9 title is incorrect. The negative binomial model is the count of hypoglycaemic events at one year, and at 2 years. Accordingly, edit rubric for ‘outcome’ column at left. --- same remark for Table 10 for DKA events • Page 55, Table 7. Missing headers? Separate matrices? • Page 62/line 57: minor typo. (Table 2)..... costs estimated by the seemingly unrelated regressions...Also see first line on the next page --- used to • Page 63, Table 12: The matrix was not correctly rendered in the PDF document. Perhaps a landscape display might help—with only the lower-triangular entries • Page 15: Acknowledgements could be listed by function and responsibility • Page 26: Please spell out PSA=probability sensitivity analysis, here and earlier on page 10, line 265
--	---

VERSION 1 – AUTHOR RESPONSE

Reviewer: 1. Andrew J Palmer
 Institution and Country: University of Tasmania, Australia

1) Abstract results – please report CIs for incremental costs and incremental QALYs.

Response: The abstract has been amended, so that the CIs for the incremental costs and QALYs are now included in the abstract.

2) Limit reporting of all QALYs in abstract, main body of the manuscript, tables etc. to 2 decimal places (currently reported to 4 decimal places, which implies that the authors can predict QALYs to the nearest hour or so over patients' lifetime.....rather unrealistic...)

Response: We agree. All QALYs reported in the abstract and the main body of the manuscript have been reduced to 2 decimal places.

3) A more detailed description of calculation of complication-specific mortality would be appreciated – currently it seems as if only ESRD-specific mortality is modelled apart from other causes of mortality. At least report that other disease-specific mortalities are modelled (e.g. from MI, stroke, heart failure, DKA, severe hypos) and whether or not HbA1c has a direct/indirect effect on these mortalities. If the model does not take into account HbA1c's impact on these forms of mortality, this is a major weakness and will lead to a higher ICER of pump vs MDI .

Response: Thanks for pointing out this matter of lack of clarity. A sentence has been added to text on page 2 lines 100-102 clarifying that people are at risk of death from ESRD, CVD events and other causes. The subsequent sentence on page 2 lines 102-104 has been added clarifying that HbA1c has an indirect on the risk of death from these events.

All references to text are made to the text in this response and all future responses refer to the clean version of the manuscript.

4) There is a good description of the impact of switching therapies on HbA1c, but the impact of switching on hypo and ketoacidosis rates is not well described – please do so. Failure to adjust for these rates changing following treatment switch will impact on the model outcomes.

Response: Both the rates of severe hypoglycaemia and DKA are dependent upon HbA1c within the model. When someone switches, their HbA1c will change as described in the main manuscript. For someone switching from CSII to MDI, their HbA1c will increase. Given this the negative binomials will predict an increase in the rate of DKA and a decrease in incidence of severe hypoglycaemia. However, different risk functions were not used for the switchers. This was because when we attempted to include switching covariates into the negative binomial models we got implausibly large coefficients (and statistically insignificant) values for some of the switching groups. This was due to small number of events within the people who switched. We have added a shorter summary of how switching effected the incidence of DKA and severe hypoglycaemia to the main text at main text, page 4, lines 164 to 169

5) Can the authors assure the readers that the number of samples/model runs was sufficient to reach a stable ICER?

Response: Stability of the model runs was determined in terms of stability of incremental net monetary benefit at £20,000 per QALY gained.

Details on this for the number of simulated individuals and the number of PSA runs have been added to supplementary material B, page 1.

Text making it clear where to find details on the stability of results have been added to the main text for 1) the stability of PSA results is on page 5, lines 224-227, and 2) the stability of deterministic results is on page 3, lines 109-110

6) While the equations for modelling HbA1c are well described, it would make the paper a lot more transparent to the non-statistician if the authors show a trace of mean HbA1c over time for each comparator. Please provide this as well as for any other biomarkers simulated in the model.

Response: We agree that adding this Figure to the paper would greatly help the readers. The trace of HbA1c over time has been provided for the deterministic base case in the main manuscript as a new Figure 1 in the HbA1c subsection of the clinical results. No other biomarkers were modelled, so only this graph was included.

7) Was there any differential impact on the comparators on BMI/weight? If so, how was this accounted for?

Response: There was no differential effect of the comparators on BMI. As such it wasn't accounted for in the model

Reviewer: 2: Anil Gumber

Institution and Country: Faculty of Health and Wellbeing, Sheffield Hallam University

1) I have noticed at several places some grammar issues...in some places interpretation were confusing. See file attached (Gumber-comments.pdf)

Response: We thank Dr Gumber for their thoroughness. We have dealt with the comments made in this pdf file in separate responses below.

2) a) It should be DAFNE+Pumps b) This should be DAFNE+MDI

Response: The abbreviations have not been changed. The preceding text in the abstract has been changed to be in the same order as the abbreviations

3) Text "A probabilistic sensitivity analysis was performed in the base" comment: on

Response: This text has been amended in line with your suggestion

4) Text "A probabilistic sensitivity analysis was performed in the base case. Further uncertainties in the cost of pumps..." Comment: "???" it should be related to sensitivity analysis of the cost of pumps

Response: We disagree that this is unclear. This text is stating that further uncertainties were explored other than the uncertainties which were included in the previously mentioned probabilistic sensitivity analysis.

5) Text "The probability of pump+DAFNE being cost-effective using a cost-effectiveness threshold of £20,000 per QALY gained was 14.0%. All scenario and subgroup analyses examined indicated that the ICER was unlikely to fall below £30,000 per QALY gained." Comment: "???" this means not significant as it falls above the threshold....very confusing with previous sentence.

Response: We are not clear what Dr Gumber means here. We see no confusion between the sentences, which address two thresholds used by NICE. The word "significant" implies a statistical analysis, and a more relevant term might have been "affordable in NICE terminology".

7) Text: "Our analysis of the REPOSE data suggests that routine use of pumps in adults without an immediate clinical need for a pump, as identified by NICE, would not be cost-effective" Comment: Very confusing

Response: We do not think this statement is at all confusing

8) Text "Historically, insulin was given twice a day, often as pre-mixed insulin, but such an approach imposes a rigid lifestyle on patients and makes it difficult to maintain blood glucose levels close to normal." Comment "Conventionally"

Response: We disagree with changing the word historically to conventionally. This was a historical approach, prior to the introduction of structured education programmes for people with type 1 diabetes. Structured education is recommended by NICE for all adults with type 1 diabetes within 12

months of diagnosis or at any other clinically appropriate time. [NICE, NG17, Recommendations 1.31, 1.3.2] As such, the regimen described should not be described as conventional

9) Text "supplementary material A." Comment "do you mean Appendix A"

Response: Yes supplementary material A is Appendix A. However, the journal style is to call additional files supplementary material, so we have used this terminology throughout the main text.

10) Text "Also, as REPOSE is the first study to assess the effectiveness of pumps+DAFNE, the long term evidence was based on observational studies of pumps in which the education component is likely to have been different" Comment "different from whom..."

Response: The text has been amended to make clear that the education in the observational studies is likely to have been different from the REPOSE study. In the clean version of the manuscript, the revised text is on page 8, lines 337 to 340 of the clean version of the manuscript.

11) Text "different countries identified" Comment "different or various"

Response: We have amended the text to now read "various countries identified"

12) Text: "The DAFNEplus NIHR programme grant, which will report in 2022, is developing and testing an adapted DAFNE course"

Comment: "reference"

Response: We have no referenced the URL to the NIHR list of funded Programme Grants for Applied Research projects on the NIHR website.

13) Dr Gumber has highlighted various words or phrases but with no explanation or comments

Response: Where appropriate, we have amended the text in line with Dr Gumber's concerns

14) Further, the model is based on RESPOSE trial which has only two cases of Black patients and none from Asian ethnicity. The evidence suggests that prevalence rate of diabetes is much higher in Asian and Black compared to White adults. This is a major limitation of modelling exercise and thus not at all representative of patients' ethnicity. So I don't think of publishing this paper in its current form.

Response: This comment shows a fundamental misunderstanding. Dr Gumber's comment about prevalence applies to type 2 diabetes. REPOSE was in type 1 diabetes, which is much more prevalent in the White population.

Reviewer: 3: Danny Liew

Institution and Country: Monash University, Australia

Comment: This is a nicely-written paper which examines the cost-effectiveness of insulin pumps, compared to multiple daily injections, over a lifetime time horizon, based on the results of the REPOSE randomised clinical trial.

Response: Thank you.

Comment: The authors utilise a published model, the 'Sheffield Type 1 Diabetes Policy Model' to undertake the analyses, with updating of some key inputs. I was not familiar with the model prior to this review, but read the original publication with interest.

Response: No revision required.

Comment: The results of the cost-effectiveness analyses, including those of the probabilistic sensitivity analyses (PSA), were well presented and appropriately interpreted. Key limitations were identified, and seemingly would not have altered the conclusion of the study anyway

Response: No response required

Comment: Ultimately, the study's applicability was limited to the patient sub-population represented by the REPOSE trial, and hence the extent to which this trial was reflective of the target clinical scenario was by and large the maximal extent to which the cost-effectiveness study was representative.

Factors that may have diminished the validity of the study's finding beyond this included the potential non-validity of the Sheffield model and the inputs entered into it. However, as mentioned, the Sheffield model has been peer-reviewed and key inputs and assumptions seemed appropriately justified.#

Response: We see this as increasing the applicability of REPOSE, since we excluded patients who would be deemed by NICE to require consideration for CSII.

1) In the main part of the manuscript, it would be helpful to describe in more detail how the Sheffield model associates HbA1c (and change in HbA1c) with risk of clinical outcomes, especially since this is the key mechanism through which benefits are achieved.

Response: Additional text describing how HbA1c effects the incidence of the long-term diabetic complications and deaths associated with these events has been added to page 2, lines 99 to 104 of the clean version of the revised main text. Full mathematical descriptions have not been added, as these are available in the referenced papers.

2) For the PSA, why were gamma distributions for disutilities associated with diabetes complications used? (Table 3) My understanding is that beta distributions (as was used for the baseline utility) are standard.

Response: Gamma distributions were chosen as they are bounded at 0 and are a standard distribution for utility decrements. (Briggs et al. Decision Modelling for Health Economic Evaluation. 2006. Oxford University Press. Page 108. Table 4.9.)

An exploratory analysis in which the 95% confidence intervals for disutilities using a Gamma and Beta distributions were compared. This indicated that there would be minimal differences between using Gamma and Beta distributions for the disutilities. Therefore we made no changes to the text or analyses.

3) Were disutilities for severe hypoglycaemia and diabetic ketoacidosis applied transiently or permanently? It should be the former, as these conditions are transient. If so, how long were they applied for?

Response: The disutilities for severe hypoglycaemia and diabetic ketoacidosis were applied for one year. This is because the source data collected EQ-5D measurements yearly, and controlled for the incidence of severe hypoglycaemia and diabetic ketoacidosis in this time period. A footnote has been added to Table 3 in the main text, clarifying whether the disutility associated with each complication is transient or not.

4) Deriving the multitude of disutility values and costs from disparate sources is a limitation that needs to be acknowledged. For example, the disutility for end-stage renal disease (-0.078) seems small compared to that for heart failure (-0.101).

Response: We thank the Prof Liew for raising this issue. This has been added a limitation in the main text on page 9, lines 353-357.

Reviewer: 4: JC Gardiner: Michigan State University, USA

Comment: Summary: The manuscript addresses the cost-effectiveness of insulin pumps (CSII) compared to multiple daily injections in Type1 diabetes adult patients, both provided with a structured education on dose adjustment for normal eating The

REPOSE trial of pumps with structured education compared to multiple daily injections (without the need for structured education) from the background, rationale, and primary data source for the

project. The design is cohort simulation of patients following the Sheffield Type1 Diabetes Policy Model. Clinical endpoints are HbA1c, and risks of complications from diabetes. The likelihood of switching treatment delivery is addressed in the analysis

Response: No change required in our text. We don't understand the highlighted text "(without the need for structured education)", since the referee makes it clear from the first sentence that he was aware that both arms received structured education.

1) The document in its entirety is 130+ pages in length, including two separate Supplements (A, B), a published paper and protocol of REPOSE. I presume the main article covers the first 27 pages, through Figure 1. It includes Tables 1-5. The remainder of the massive document, includes supplementary details from REPOSE, a within-trial incremental cost-effectiveness analysis. Methods employed and model assumptions are described in great detail. All are appropriate. The cost-effectiveness analyses follow generally accepted guidelines for conduct and reporting of results—of course NICE (and ISPOR) recommendations are followed.

Response: If the manuscript is accepted, then the final version will be much shorter as the published protocol paper and the full protocol will not be included. The other comments are complimentary, for which we thank the Prof Gardiner.

2) Comment Apart from the objectives of this specific study, this "project" is valuable for providing a sort of tutorial for delineating the compulsory elements for conduct, design and analysis of a costeffectiveness study that has the backdrop of a substantive clinical trial with patient level data and resource utilization. To be useful in this regard, the online document must be assembled with links to assess table, figures and supplement. Up front a table of contents, links to chapters, etc is highly recommended.

and

Comment: Format the article as a mini-book, with index of contents, chapters and links to each chapter. I guess the Supplementary Materials A and B will be separate documents. (eg, The REPOSE Protocol from page 72)

Response: We note that editor's comment that we should not take up the referee's suggestion that we convert the paper to a book. This is unnecessary since some of the submitted background documents need not be attached to the final version of 27 or so pages.

3) After page 42 'pumps' is replaced by CSII=continuous subcutaneous insulin injection

Response: Thank you for pointing this out, we have amended the supplementary materials so that they refer to pumps rather than CSII

4) It would help the reader if a list of all abbreviations and acronyms placed upfront, perhaps just before the INTRODUCTION. At first mention later in the article the acronym is spelled-out and afterwards may be maintained.

Response: A list of abbreviations is not given in the format of the articles that this journal publishes, so this has not been added to the main text. However, we have added this table to each of the supplementary materials.

5) Please give a reference to SIMUL8 (2010) software, and for STATA

Response: These references have been added to the main text. In the spirit of your comment, a reference to the R software used has also been added to the main text.

6) Hard to locate Supplements A and B

Response: Supplementary materials A and B have now been given clear titles within their texts.

7) Cohort simulation Groot Koerkamp et al [16]. Please see if Groot Koerkamp et al (Medical Decision Making, JUL-AUG 2011) is a more appropriate reference.

Response: Thank you for pointing this out, the reference has been changed

8) Page7/line 125; ?replicated

Response: Thank you for pointing this out. Replicates has been changed to replicated

9) Sentences 126-130 require clarification....[?] –single imputation was performed ..., by two procedures...(i) chained regression....., (ii) Poisson... What does ‘both sets’ of imputation models refer to?

Response: The description of the simulation cohort in the main text has been rewritten, aiming to address the lack of clarity in this section. Both sets of imputation models was meant to mean both the truncated chained regression and Poisson imputation procedures. In the main manuscript the description of how the simulation cohort was generated is now given in the main text on page 3 lines 129-144.

10)Page 7/line 136: suggest sub-rubrics for the three models that follow for (a) treatment switching, (b) HbA1c, (c) severe hypoglycaemia and DKA. They are separate outcomes in the model that are discussed next

Response: These sub-rubrics have been added to the results section of the main paper

11) Page7/line 143 ff: Time to treatment-switching as event. In Supplementary Materials B.

Suggest ----survival curves estimated from parametric models were plotted against the nonparametric Kaplan-Meier curve. There is only one KM curve (Figure 1/page 49 and Figure 2/page 50) with 5 comparison parametric survival curves obtained from the Exponential, Weibull, Gompertz, log-logistic and log-normal distributions. For each these some sort of mixture distribution must be used because there are covariates in the parametric model (Tables 1-2, page 45-47)

--- Is HbA1c (as covariate) in these model the single baseline value? In other words, it is not time-varying. HbA1c is recorded at baseline, 1 year and 2 years. However, many more assessments are likely available in this study.

--- Only available covariate data were used in the time-to-event (treatment switching) estimations. So, mention to operational sample size (n=xxx) in the legend on the tables

Response: a)The text in Supplementary material B has been amended to reflect this suggested wording on page 2 paragraph 5

b)The functional form of all S(t) functions are now given within tables 1 and 2 of supplementary material B. This includes all of the covariates given in these tables to estimate S(t).

c) As mentioned in Supplementary material B, pg 2, paragraph 2, the HbA1c covariate was HbA1c prior to switching or 2 years follow up if no switching occurred. Likewise the number of severe hypoglycaemic events and number of DKAs in the year prior to switching or year prior to the two year follow up were included

d)The operational sample size has been added to the table legends of Tables 1 and 2 of supplementary material B.

12) Table 1-2 (pages 45-47): Presumably these are output by STATA. The shape/scale parameter names such “gamma”, “sigma” parameters, could be confusing unless we know the parameterization used. The survival functions (not all) are of the form $S(t) = S_0((t/\theta)^\gamma)$ for scale θ and shape γ . For goodness-of-fit, criteria such as Kolmogorov-Smirnov (KS), Anderson-Darling (AD), Cramer von-Mises (CvM) should be added to Table 3. Where statistics involve the log-likelihood, $-2 \log L$ should include all estimated constants and based on the un-logged response.

Response: a) The tables have been clarified. The how to calculate the survivor function (S(t)) used by each of the models have been added in Tables 1 and 2.

b) The $-2 \times \log$ p-values likelihood has been added to Table 3 for all models. This includes all estimated constants and is based on un-logged response (default reporting of this figure in STATA).

c) In health technology assessments, by UK decision makers, the Kolmogorov-Smirnov (KS), Anderson-Darling (AD), Cramer von-Mises (CvM) are not standard tests used to assess the goodness of fit of parametric survival models to the underlying Kaplan-Meier data. (Latimer 2011) Furthermore, we believe that you are asking us to test whether the martingale survival residuals (which are bounded between -0.5 and +0.5) are equivalent to a uniform distribution between -0.5 and +0.5. (Price and Jones 2017). The martingale survival residuals cannot be obtained with a default command in the STATA software that we used to fit the curves. Only the standard martingale residuals (bounded between $-\infty$ to 1) are available from STATA. Hence, we do not think that we are able to easily perform the requested statistical tests.

References

Latimer, N. NICE DSU Technical Support Document 14: Undertaking survival analysis for economic evaluations alongside clinical trials - extrapolation with patient-level data. 2011. Available from <http://www.nicedsu.org.uk>

Price, M.H., Jones, J.H., A general goodness-of-fit test for survival analysis. bioRxiv preprint. Uploaded Jan. 31, 2017. Last Accessed 28th November 2017. doi:<http://dx.doi.org/10.1101/104406>.

13) Page 51, Table 3 begs an explanation. Rubrics for row/column misaligned

Response: The CSII+DAFNE and MDI+ DAFNE titles have been moved to be across the relevant rows rather than across the columns of table 3 in Supplementary material B. An explanation of how the information presented in table 3 was used to determine the base case curves is provided in Supplementary material B, page 3, paragraph 5.

14) Comment: Back to Page 8/para 1: This is a model for HbA1c to aid in imputation. So, drop "in the model"

Response: This part of the sentence has been removed

15) Comment: Page 8 line 16: supplementary material (pg8) is actually pg9. In Supplementary Material: on this page (page 52), the beta regression model uses the logit link for the mean $E(Y|x) = \mu(x)$ and the variance is $\text{Var}(Y|x) = \phi\mu(x)(1-\mu(x))$. The dispersion is on the variance (not on the mean as stated). Some software use $(1+\phi)^{-1}$ in place of ϕ .

In tables that follow,

--- Does "beta scale" mean "logit scale"?

--- Natural logarithm of phi. I guess it is the parameterization above

--- Patients were nested in one of 7 clinic sites. Any concerns or issues on analyses if clustering should be accounted for?

Response: a) Thank you for pointing out the text stated "dispersion on the mean", this has been changed throughout.

b) Beta scale meant HbA1c was estimated with 0 = 29 mmol/mol and 1 = 201 mmol/mol. An explanation of this has been added as a footnote to Supplementary Material B Tables 5 and 6.

c) To make Supplementary Material B Tables 5 and 6 to make clearer the description of the mean effect and dispersion parameters have been changed to "Mean effect (Mu) – using a logit link function" and "Dispersion parameter (phi) – using a natural logarithm link function". Respectively

d) Clustering in the each delivered course, rather than each site, was controlled for in these analyses. This detail is given in supplementary material B, pg10, paragraph 1. As the trial was not randomised in clusters to CSII+DAFNE or MDI+DAFNE centres, this was the most appropriate design. Centre effects were covariate adjusted and this is in line with the statistical analysis plan used to analyse the trial data.

16) Comment: "Page 56 (pg13 in Supplement): Is Table 9 to correct reference?"

Response: Thank you for pointing this out. The table reference should have been Table 8 and this has been amended.

17)Comment: Page 59 (pg16 in Supplement): Table 9 title is incorrect. The negative binomial model is the count of hypoglycaemic events at one year, and at 2 years. Accordingly, edit rubric for 'outcome' column at left. --- same remark for Table 10 for DKA events

Response: Thanks for pointing these things out. The titles of tables 9 and 10 of supplementary material B have been amended to remove the baseline.

18) Comment: Page 55, Table 7. Missing headers? Separate matrices?

Response: Tables 4,7 and 12 were in an embedded excel workbook containing the variance-covariance matrices for all fitted models. This is what caused the issues with rendering the PDF. All variance-covariance matrices used to parameterise the economic model have now been added as separate tables to supplementary material B.

19) Comment: Page 62/line 57: minor typo. (Table 2)..... costs estimated by the seemingly unrelated regressions...Also see first line on the next page --- used to

Response: The header of Table 2 has been amended. Thank you for pointing out these typos, they have been amended.

20)Comment: Page 63, Table 12: The matrix was not correctly rendered in the PDF document.

Perhaps a landscape display might help—with only the lower-triangular entries

Response: See our response to your 18th comment

21) Page 15: Acknowledgements could be listed by function and responsibility

Response: The contributions of the authors have been amended into this format. The acknowledgements for the rest of the REPOSE group have been left in the old format for consistency with other papers from the REPOSE trial.

22) Page 26: Please spell out PSA=probability sensitivity analysis, here and earlier on page 10, line 265

Response: Thanks for pointing this out. PSA has been added as an acronym when it is first mentioned and the legend of figure 1 has been amended so that it says probabilistic sensitivity analysis.

VERSION 2 – REVIEW

REVIEWER	Danny Liew Monash University, Australia
REVIEW RETURNED	09-Dec-2017

GENERAL COMMENTS	The authors have adequately addressed my initial comments.
--

REVIEWER	JC Gardiner Michigan State University, USA
REVIEW RETURNED	22-Dec-2017

GENERAL COMMENTS	The authors have taken seriously the comments and suggestions
---

	made by the reviewers. It is still a massive document and whether or not the final version will end up as a book with links to the various materials is a decision that would be taken at the editorial office. Supplementary Material A and B have also been revised/corrected and links provided to the sub-chapters. This is enormously helpful. Similarly, and for example, (with comma) is a style that that authors might consider. On page 11, NovoNordisk appears in three forms. Figure 1 and Figure 2 do not show up on the DOC version that I downloaded I have no additional comments.
--	---